# The Need for the Optimization of HIV Antiretroviral Therapy in Kazakhstan

**DOI:** 10.3390/v17050690

**Published:** 2025-05-10

**Authors:** Aidana Mustafa, Natalya Dzissyuk, Bauyrzhan Bayserkin, Dinara Begimbetova, Zhamilya Nugmanova, Syed Ali

**Affiliations:** 1Department of Biomedical Sciences, School of Medicine, Nazarbayev University, Astana 010000, Kazakhstan; aidana.mustafa@nu.edu.kz; 2Kazakh Scientific Center of Dermatology and Infectious Diseases, Almaty 050094, Kazakhstan; lab4@kncdiz.kz (N.D.); bbaiserkin@gmail.com (B.B.); 3Laboratory of Molecular Oncology, Center for Life Sciences, National Laboratory Astana, Nazarbayev University, Astana 010000, Kazakhstan; dinara.begimbetova@nu.edu.kz; 4HIV Infection and Infection Control Division, Department of Epidemiology, Asfendiyarov Kazakh National Medical University, Almaty 050094, Kazakhstan; zhamilya.nugmanova@gmail.com

**Keywords:** HIV, ART, DRM, drug resistance, Kazakhstan

## Abstract

The number of people living with HIV in Kazakhstan increased from 11,000 to 35,000 between 2010 and 2021, with emerging antiretroviral therapy (ART) resistance posing a challenge to effective treatment. Unsafe injection practices among people who inject drugs (PWID), the stigma against men who have sex with men, sex work, drug possession, HIV transmission, HIV exposure, and the non-disclosure of HIV status create obstacles to effective prevention and care. Our recent studies with people living with HIV (PLWH) in Kazakhstan have revealed the prevalence of mutations in HIV that may confer resistance to certain ART components currently being administered in the country. Additionally, subtype A6- and CRF02_AG-infected PLWH displayed the occurrence of certain distinct subtype-specific DRMs. Subtype A6 exhibited a tendency for the DRMs A62V, G190S, K101E, D67N, and V77I, whereas CRF02_AG was more associated with S162A, K103N, and V179E. Both subtypes had a comparable frequency of the M184V mutation and displayed similar patterns in the distribution of Q174K. Based on our findings, we recommend that DRM screening and subtype diagnosis before the initiation of ART will improve treatment efficiency while preventing the emergence of further DRMs in Kazakhstan.

Despite a globally reported 38% decline in the incidence of HIV infections between 2010 and 2022, during the same period a 49% increase has been reported in Eastern Europe and Central Asia [1]. Notably, the Republic of Kazakhstan, one of the largest countries in Central Asia by population and land area, experienced a significant 73% rise in HIV incidence from 2010 to 2020. During the same period, the number of people living with HIV (PLWH) surged from 11,000 in 2010 to 35,000 by 2021 [2,3]. Additionally, a two-fold increase in HIV prevalence is anticipated over the next decade [4].

The rising incidence of HIV in Kazakhstan can be attributed to multiple factors, primarily the inadequate coverage of prevention and treatment services. As of 2025, 83% of people living with HIV (PLHIV) are aware of their status, 90% are receiving treatment representing only 74.7% of all estimated PLHIV, and 92% of those on treatment have achieved viral suppression, covering just 68% of all PLHIV [3,5]. Overall, the 92% viral load suppression rate demonstrates the effectiveness of Kazakhstan’s recommended first-line regimen. However, further optimization is needed to better align with the region’s specific epidemic dynamics. Given the distinct HIV subtypes and variants prevalent in Central Asia (primarily A6 and CRF02_AG), a tailored treatment approach that accounts for these unique epidemiological factors is essential [6].

Currently, viral load testing is available at all HIV centers in Kazakhstan. Patients diagnosed with HIV are promptly offered antiretroviral therapy (ART), which is readily accessible. However, treatment coverage remains a challenge due to several socioeconomic and structural challenges which further exacerbate the HIV epidemic in the country. Unsafe injection practices among people who inject drugs (PWID) persist, largely due to the limited availability of opioid agonist therapy. Legal and policy barriers, including the criminalization of sex work, drug possession, HIV transmission, HIV exposure, and the non-disclosure of HIV status, create additional obstacles to effective prevention and care. Social stigma surrounding HIV and the widespread stigma and discrimination against gay men and men who have sex with men (MSM) continue to hinder access to essential healthcare services [1].

To improve HIV treatment coverage in Kazakhstan, healthcare services should be decentralized to increase accessibility, especially in rural areas. Mobile health units can reach underserved populations, while awareness campaigns can help reduce stigma and encourage testing. Integrating HIV services with other healthcare programs, such as maternal health and TB care, can streamline access to treatment. Expanding telemedicine and digital health solutions will provide remote consultations and adherence support. Increased government funding and policy reforms can help lower the cost of antiretroviral therapy. Strengthening peer support networks and distributing HIV self-testing kits will further encourage early diagnosis and treatment. Lastly, improving data monitoring systems will help track progress and optimize resource allocation.

Kazakhstan has made significant progress in updating its HIV treatment protocols to align with international standards. In 2020, the Ministry of Health authorized a clinical protocol recommending a preferred regimen combining two nucleoside reverse transcriptase inhibitors (NRTIs) with integrase inhibitors (INIs), and alternative regimens consisting of two NRTIs and one non-nucleoside reverse transcriptase inhibitor (NNRTI). By 2023, the country integrated dolutegravir-based regimens as the preferred first-line treatment, reflecting a commitment to modern, effective therapies [7]. However, challenges remain in fully transitioning from older antiretroviral therapy (ART) regimens. Studies by our team have identified resistance mutations associated with drugs like zidovudine and nevirapine, which were more prevalent in earlier treatment protocols. The continued use of earlier protocols may stem from factors such as the high cost of newer medications, bureaucratic challenges in updating national guidelines, and a lack of widespread training for healthcare providers. Additionally, some clinicians may be hesitant to prescribe newer regimens due to limited experience, concerns about potential side effects, or uncertainty regarding their long-term efficacy. Strengthening policy support, improving clinician education, and ensuring the affordability of updated treatments are essential steps in facilitating a full transition to modern HIV therapies [7].

Following WHO recommendations and European AIDS Clinical Society guidelines, the Ministry of Health in Kazakhstan has approved a clinical protocol for HIV treatment that mandates universal ART administration within 7 days of diagnosis, irrespective of the clinical stage or CD4+ cell count [8]. The first-line preferred management includes the combination of NRTIs such as lamivudine (3TC) or emtricitabine (FTC) with tenofovir (TDF), or NRTI/INSTI combinations like tenofovir alafenamide (TAF) with dolutegravir (DTG), 3TC or FTC/TDF or TAF with bictegravir (BIC), and 3TC/abacavir (ABC) with DTG. First-line alternative regimens include NRTI/NNRTI combinations like 3TC or FTC/TDF with efavirenz (EFV), or NRTI/INSTI combinations such as TDF with raltegravir (RAL). Other alternatives are NRTI/NNRTI combinations like TAF with doravirine (DOR), NRTI/PI combinations like ABC with darunavir-cobicistat (DRV/c) or darunavir-ritonavir (DRV/r), and NRTI/NNRTI combinations like TAF with rilpivirine (RPV). Additionally, NRTI/INSTI options include 3TC with dolutegravir (DTG), and NRTI/NNRTI combinations like cabotegravir (CAB) with rilpivirine (RPV) [8].

An additional obstacle to effective antiretroviral therapy (ART) is the emergence and spread of HIV drug resistance [9]. The development of drug-resistant HIV is promoted by several factors, including prolonged viral replication in the presence of subtherapeutic levels of antiretrovirals (ARVs), the impact of drug resistance mutations (DRMs) on drug efficacy and viral replication, and the ease with which specific DRMs can be acquired [10]. Drug resistance can develop in two main ways: through the emergence of DRMs in individuals receiving non-suppressive ART or via transmission from individuals infected with drug-resistant HIV strains [11].

In Kazakhstan, national surveys on HIV drug resistance have not been conducted since the adoption of updated WHO-recommended ART guidelines. Earlier studies, however, provide insight into the prevalence of HIV drug resistance. Among therapy-naïve PLHIV studied between 2009 and 2013, the prevalence of drug resistance was relatively low at 3% [12]. In contrast, our recent analysis of HIV drug resistance mutations (DRMs) among 602 Kazakhstani PLWH reported a high prevalence of DRMs across all three regions (PIs, NRTIs, and NNRTIs)—32%—among PLHIV receiving ART [7]. These findings underscore the urgent need for the routine monitoring and surveillance of HIV drug resistance in Kazakhstan to inform effective treatment strategies and mitigate the growing threat of resistance.

The emergence of ART resistance poses a significant challenge to the effective management of HIV infection. A study indicated that in Kazakhstan, the viral load suppression (VLS) rate among patients on ART was 92%, based on the data showing that 90% of individuals aware of their HIV-positive status were receiving treatment [5]. In contrast, Tajikistan, Kyrgyzstan, and Ukraine reported VLS rates of, respectively, 87%, 91%, and 95% [13,14,15]. Our analysis of HIV drug resistance mutations (DRMs) among 602 Kazakhstani PLWH showed that only 37.5% of the study participants were receiving the ART regimen recommended by the Ministry of Health, while 32% exhibited DRMs [7]. Among the study participants, 32% were receiving the alternative combination regimen of TDF/FTC/EFV. In contrast, 20%, 15%, and 1.5% were on the regimens ZDV/3TC/NVP, ZDV/3TC/EFV, and TDF/FTC/NVP, respectively—none of which were classified as either preferable or alternative options. Despite global shifts in treatment recommendations, D4T and ZDV continue to be widely used in first-line regimens across many Asian countries. Although, due to concerns about their safety and efficacy, these drugs are no longer recommended as first-line therapies in high-income nations, their affordability makes them a more viable option in resource-limited settings [16,17,18]. This reliance on older and less optimal regimens highlights the persistent inequities in access to newer antiretroviral therapies, which could impact treatment outcomes and long-term efforts to control the HIV epidemic in the region.

Several factors hinder the implementation of proper treatment with first-line ART in Kazakhstan, including regulatory or policy barriers, limited access to medical services, and medication shortages, particularly in rural areas. Additionally, social stigma surrounding HIV/AIDS remains a significant challenge. Among people who inject drugs (PWIDs), adherence to ART is further complicated by substance use, the lack of social support, and structural barriers such as stigma and transportation difficulties. These factors collectively undermine the effectiveness of ART, contributing to the emergence and spread of drug-resistant HIV strains [19,20,21]. Overcoming these obstacles requires a comprehensive approach, including policy reforms to improve access to affordable medications, investment in healthcare infrastructure, public education campaigns to reduce stigma, and targeted interventions to support adherence among key populations. Implementing these strategies would facilitate the successful adoption of appropriate first-line ART in Kazakhstan.

Previously, we found that the most common drug resistance mutations (DRMs) observed were the NRTI accessory mutation A62V (22%), the NNRTI major mutation K103N (17%), and the NRTI major mutation M184V (8%) [7]. The A62V mutation is linked to multi-resistance to NRTIs, while K103N is associated with high-level resistance to nevirapine (NVP) and varying levels of resistance to efavirenz (EFV). The M184V mutation is primarily associated with high-level resistance to lamivudine (3TC). The occurrence of these mutations highlights the complexity of resistance patterns and their implications for treatment efficacy [7]. Interestingly, DRM distribution exhibited a subtype-based variation. The DRM A62V was identified in 132 (39%) patients infected with the A6 subtype and in only 8 (3%) infected with CRF02_AG. The V77I mutation was more frequently observed in A6 subtype patients (26%), compared to 1.6% among the patients infected with CRF02_AG. In contrast, the DRMs S162A and K103N were more frequently observed among CRF02_AG-infected PLWH (94% and 40%, respectively) than among those infected with subtype A6 (2% and 15%, respectively). Both subtypes displayed an even distribution of the DRMs M184V and Q174K [7].

The distribution of antiretroviral therapy (ART) resistance among participants in this study was as follows: 11 individuals (2%) exhibited resistance to protease inhibitors (PIs), 21 (3%) to nucleoside reverse transcriptase inhibitors (NRTIs), and 108 (18%) to non-nucleoside reverse transcriptase inhibitors (NNRTIs). Additionally, resistance to multiple drug classes was observed: 1 participant (0.2%) showed resistance to both PIs and NRTIs, 3 (0.4%) to both PIs and NNRTIs, and 47 (8%) to both NRTIs and NNRTIs. Thus, fewer participants living with HIV (PLHIV) were resistant to protease inhibitors (PIs) compared to nucleoside reverse transcriptase inhibitors (NRTIs) or non-nucleoside reverse transcriptase inhibitors (NNRTIs). Specifically, three participants (0.5%) exhibited high-level resistance to fosamprenavir (FPV/r), two (0.3%) to nelfinavir (NFV), and one (0.2%) each to atazanavir (ATV/r), indinavir (IDV/r), lopinavir (LPV/r), and saquinavir (SQV/r). In terms of NRTI resistance, 59 participants (10%) had high-level resistance to emtricitabine (FTC) or lamivudine (3TC). Resistance to other NRTIs was less common: 15 (3%) to didanosine (DDI), 13 (2.2%) to abacavir (ABC), 6 (1%) to stavudine (D4T), 4 (0.7%) to zidovudine (ZDV), and 4 (0.7%) to tenofovir (TDF). For NNRTIs, high-level resistance was observed as follows: nevirapine (NVP) in 21%, efavirenz (EFV) in 20%, rilpivirine (RPV) in 4%, doravirine (DOR) in 1.7%, and etravirine (ETR) in 1%. Further, our study revealed a statistically significant decrease in CD4 counts among PLHIV with the reverse transcriptase (RT) ARV resistance mutations (ARTRMs) E138A (accessory), K103N (major), and Q174K (polymorphic other) compared to those without these mutations (*p*-value < 0.05). Furthermore, PLHIV with the Q174K mutation exhibited significantly higher viral loads compared to those without this ARTRM, suggesting the potential impact of these mutations on both immune function and viral control.

The odds of E138A polymorphic mutation occurrence in the viral RT region were found to be increased with the use of ZDV or NVP (Figure 1A). Interestingly, the simultaneous administration of these drugs was negatively correlated with the presence of E138A, suggesting that the use of ART combinations containing both ZDV and NVP might potentially be beneficial in reducing the odds of E138A emergence. Another polymorphic accessory protease inhibitor (PI) mutation, L10V, was found to be positively associated with the administration of DRV, a protease inhibitor. Similarly, the occurrence of the RT mutation S162A was found to be positively correlated with the administration of NVP and with CRF02_AG infection (Figure 1A) [7].

Considering the outcomes of this study, namely, the presence of the K103N mutation (17%), known to confer resistance to efavirenz, a component of the alternative first-line regimen (3TC or FTC/TDF/EFV) in Kazakhstan, and the M184V mutation (8%), recognized for inducing high-level resistance to 3TC, that is included in almost all preferred and alternative first-line regimens, we recommend replacing EFV and 3TC in the currently prescribed cocktails. One of the primary drawbacks of efavirenz (EFV) is its tendency to induce single amino acid mutations, such as K103N and V106M, which can result in high-level drug resistance [22,23,24]. The development of DRMs contributes to a reduced virological response, which in turn increases the risk of transmitting the mutated virus to newly infected individuals [25,26,27]—increasing the likelihood of developing multidrug resistance in drug-naïve PLWH [28]. Given these risks, it is strongly recommended to conduct drug resistance testing before initiating antiretroviral therapy to guide more effective treatment decisions and reduce the potential for resistance development. Additionally, our study revealed a negative association between the simultaneous administration of ZDV with NVP and the presence of E138A, indicating a potential benefit in combining the two drugs [7].

Another study conducted by our team, analyzing 968 HIV-1 sequences obtained from PLHIV in Kazakhstan, unveiled a notable prevalence of high-level drug resistance (32.1%) [6]. More specifically, an increase in high-level resistance to NRTIs and NNRTIs was observed, reaching 16.8% and 30%, respectively. Furthermore, this study reported a higher prevalence of drug resistance in individuals prescribed with second-generation (aOR: 2.74, 95% CI: 1.83–4.16) and mixed-generation (aOR: 1.53, 95% CI: 1.00–2.37) ART regimens compared to those on first-generation ART regimens. Second-generation ART regimens typically include newer, more potent antiretroviral drugs from various classes (PIs, NRTIs, NNRTIs, INSTIs, etc.) that were developed after the first generation to have improved resistance profiles, fewer side effects, and better efficacy, while mixed-generation ART regimens combine drugs from both first- and second-generation ART [29,30]. Additionally, our study identified a higher frequency of A62V, G190S, K101E, and D67N in subtype A6, while K103N and V179E were more prevalent in CRF02_AG (Figure 1B). The mutation K103N was found to be the most frequent major NNRTI-associated DRM in subtype A6-infecetd patients, potentially conferring resistance to NVP and EFV [6]. The occurrence of subtype-specific unique DRMs highlights the need for subtype diagnosis prior to the prescription of ART.

A recent analysis from our group examined the transmission of high-level HIV-1 subtype A6 drug-resistant mutations across the 14 former Soviet Union (FSU) countries, revealing a significant proportion of ART-naïve individuals with high levels of resistance to NRTIs and NNRTIs—1.2% and 2.8%, respectively [7]. These findings likely indicate transmitted resistance, emphasizing the importance of conducting drug resistance mutation testing early in the course of infection, ideally before initiating ART. Such testing can help guide the selection of effective treatment regimens right from the start. Furthermore, enhanced surveillance, robust ART adherence programs, and expanded prevention efforts are essential for reducing the burden of resistance and preventing the spread of multidrug-resistant strains. Considering the history of cross-border population mobility and the interlinked HIV epidemics within the FSU region, the emergence of similar DRMs in other FSU countries is highly likely [31,32,33,34]. Hence, our findings can inform further research and the development of clinical guidelines not only for Kazakhstan but also for other FSU countries.

In summary, our analyses highlight a significant prevalence of drug resistance among PLHIV in Kazakhstan, emphasizing the necessity for DRM screening prior to initiating ART treatment. Since PLWH carrying a high viral load are more likely to transmit the infection, it is imperative to optimize the ART regimen for these populations. Conducting a nationwide study to detect DRMs among PLWH is also recommended. The novelty of this study lies in its comprehensive approach, considering not only the prevalence of DRMs but also the diversity of HIV subtypes and the distinct characteristics of the epidemic in Central Asia. Given the predominance of subtypes A6 and CRF02_AG in the region, optimizing treatment strategies to align with these epidemiological factors is essential for improving efficacy and mitigating the risk of emerging drug resistance mutations. The diverse frequency of DRMs across HIV subtypes underscores the importance of testing for the subtype to optimize the initiation of ART. Integrating these recommendations into clinical guidelines for HIV management in Kazakhstan will enhance the efficiency of ART while mitigating the occurrence of new DRMs, and will accelerate the progress towards achieving the 90-90-90 ART cascade target (UNAIDS.org).

## Figures and Tables

**Figure 1 viruses-17-00690-f001:**
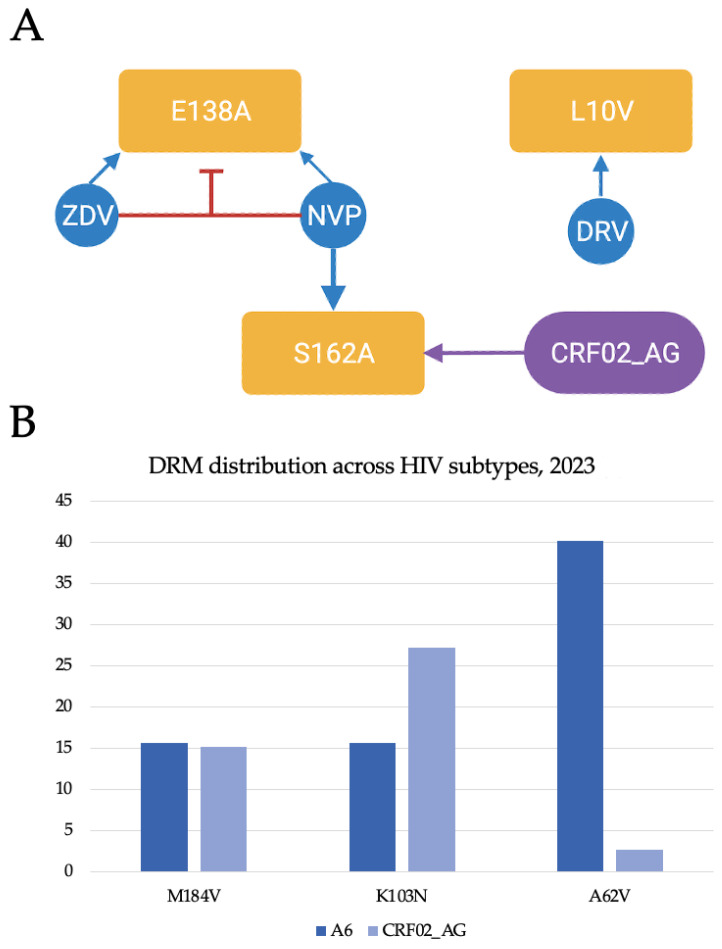
(**A**) Statistically significant associations between DRMs, ART components, and subtypes from our analysis of HIV DRMs among 602 Kazakhstani PLWH [10]. (**B**) Distribution of most frequent DRMs by subtypes A6 and CRF02_AG in Kazakhstan [11].

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
