# Peer review of "The Need for the Optimization of HIV Antiretroviral Therapy in Kazakhstan"

_viruses, 2025, doi:10.3390/v17050690_

Round 1

Reviewer 1 Report

Comments and Suggestions for Authors

The Authors’ commentary  focuses  on the need to optimize HIV care in Kazakhstan, based on the increased number of PWH over the last decade 2010-21 (73% increase) in this Country.

As the abstract suggests, they primarily focus on the DRM prevalence and the need to implement HIV resistance testing in their regions.

Despite being a reasonable request to improve the quality of HIV care, the data reported by the Authors suggest additional considerations from the Public Health perspectives (as the commentary suggest) which deserve to be analysed and possibly influence the structure of the commentary.

In fact, they report on the low treatment coverage  (and treatment cascade, compared to the UNAIDS 95-95-95% target), suggesting that only 48% of PWH receive therapy. So, the call for action should be first to analyse the reasons of such undercover and suggest possible intervention strategies to overcome the current barriers.

Similarly, as the Ministry of Health has supported the use of 2-generation INSTI-based regimens with rapid initiation: < 7 days (outstanding also for European Countries), the reasons for persistent use of “old regimens” require to be investigated (is there a recommendation to systematic roll over the treatments? which are the reasons for not switching (financial, organizational, fear to change therapy for PWH, or insufficient confidence by clinicians). Is viral load testing available for all HIV centres? Does social stigma play a relevant role? How many are rapidly taken into care? The discrepancy between health authorities mandates and real life need to be

The issue of DRMs (in the era of 2 generation INSTI regimens with high barrier to resistance) becomes relevant in the context of optimization of  therapy in treated PWH, but it is not expected per sè to increase treatment coverage, which appear to be the main issue at the moment. The optimization is reasonably a second step to improve the standard of care.

So, I think this reasonable and welcome call for action should be shaped differently and provide proposals in order to  get closer the UNAIDS target and not be limited to the implementations of resistance testings alone (as it appears from the abstract), which would leave all the other issues unanswered. The lesson from African Countries (see Botswana) do confirm the strategy is possible even in resource-limited areas. So I would suggest to interpret the optimization in a Public Health perspective

Author Response

Please see response in attached file.

Reviewer 2 Report

Comments and Suggestions for Authors

This manuscript is a well-documented review of recent drug-resistant HIV in Kazakhstan.

Abstract

Please provide a little more information about the recent HIV drug-resistant mutations (DRMs) in Kazakhstan in the abstract to make it easier for readers to understand the current situation in Kazakhstan.

Main text

1. Please update the global and Kazakhstan’s HIV epidemiology based on the latest Global HIV Epidemiology Report published in 2024.

2. What are the reasons why, contrary to the global trend, the number of people living with HIV (PLWH) increased from 2010 to 2021 and why is the HIV prevalence expected to increase in the next decades in Kazakhstan? Please consider to adding explanation for them in the text.

3. The current PEPFAR website doesn’t provide information on Kazakhstan. These references must be updated to appropriate ones.

4. What does it mean “viral coverage” in pages 2.

A study indicated that in Kazakhstan, the viral load suppression (VLS) rate among patients on ART was 89.48%, based on a viral coverage of 99.46% [12].

5. The reviewer strongly agrees with the nationwide surveillance of HIV DRMs among ART-naïve as well as the monitoring of HIV DRMs on ART. However, the first-line ART regimens currently recommended by the Ministry of Health in Kazakhstan are effective against the HIV DRMs currently prevalent in Kazakhstan. Therefore, the reviewer believes that it is important to implement the appropriate first-line ART in Kazakhstan at the same time. What barriers prevent the use of proper first-line ART in Kazakhstan? Please consider adding discussions on this issue to the text.

6. What do they mean "second-generation” and” mixed-generation” in page 5?

Furthermore, this study reported a higher prevalence of drug resistance in individuals prescribed with second-generation (aOR: 2.74, 95% CI: 1.83–4.16) and mixed (aOR: 1.53, 95% CI: 1.00–2.37) generation ART regimen compared to those on first-generation ART regimen.

7. Please confirm all references are accessible and applicable as of the submission of this manuscript. If the references are not applicable, please update them with relevant ones.

Author Response

Please see response in attached file.

Reviewer 3 Report

Comments and Suggestions for Authors

The manuscript by Mustafa et al describes the number of people living with HIV in Kazakhstan increased from 11,000 to 35,000 between 2010-2021, with emerging antiretroviral therapy (ART) resistance.  Their study revealed prevalence mutation in HIV that may confer resistance to certain ART currently administered.  Based on their findings they recommend that DRM screening and subtype diagnosis before the initiation of ART. The study is straight forward.  However, prevalence of drug resistance and their recommendation have been described several times in the past.  It is difficult to gauge the new contribution in the current manuscript except the study is conducted in one geographical location.  The authors should convincingly establish the novelty of their study to warrant publication.

Other points

  1. What is the evidence that each mutation they described exactly correlate with drug resistance except pointing out prevalence.  Have they sequenced other regions that may confer and contribute to drug resistance?

  1. Paragraph 3: The authors should include the target regions for each drug mentioned so that readers can easily follow understand the content of the manuscript.

Author Response

Please see response in attached file.

Round 2

Reviewer 1 Report

Comments and Suggestions for Authors

Few additional adjustments are needed for a higher clarity:

line 136: in which regions (NRTI? NNRTI? PI?)

line 111-119: the list of options needs rewording, as it appears difficult to read . Some spelling mistakes for ARVs

line 121: I would write "additional"

line 142: "ART coverage 99%" is still unclear to me.

line 173:  it needs reference (our analysis?). if this is here presented, then it requires a different narrative: "we have investigated ..... etc"

line 231: the standard acronym is DRMs (drug resistance mutations)

line 234: use only "PLWH" as short , as it is has already been mentioned above

line 241-44: rewording to avoid repetition

line 250: atypical reporting. usually it is: 2nd generation INSTI (i.e. DTG or BIC) ....

Reviewer 3 Report

Comments and Suggestions for Authors

Comments are addressed adequately.

Author Response

All comments were addressed in the previous revision.